# A Review of Decontamination *of Aspergillus* spp. and Aflatoxin Control for Grains and Nuts with Atmospheric Cold Plasma

**DOI:** 10.3390/toxins17030129

**Published:** 2025-03-10

**Authors:** Miral Javed, Wei Cao, Linyi Tang, Kevin M. Keener

**Affiliations:** School of Engineering, University of Guelph, 50 Stone Road East, Guelph, ON N1G 2W1, Canada; mjaved11@uoguelph.ca (M.J.); wcao02@uoguelph.ca (W.C.); ltang07@uoguelph.ca (L.T.)

**Keywords:** cold plasma, molds, mycotoxins, inactivation, detoxification, food safety

## Abstract

*Aspergillus* spp. and their produced aflatoxins are responsible for contaminating 25–30% of the global food supply, including many grains, and nuts which when consumed are detrimental to human and animal health. Despite regulatory frameworks, *Aspergillus* spp. and aflatoxin contamination is still a global challenge, especially in cereal-based matrices and their derived by-products. The methods for reducing *Aspergillus* spp. and aflatoxin contamination involve various approaches, including physical, chemical, and biological control strategies. Recently, a novel technology, atmospheric cold plasma (ACP), has emerged which can reduce mold populations and also degrade these toxins. ACP is a non-thermal technology that operates at room temperature and atmospheric pressure. It can reduce mold and toxins from grains and seeds without affecting food quality or leaving any chemical residue. ACP is the conversion of a gas, such as air, into a reactive gas. Specifically, an electrical charge is applied to the “working” gas (air) leading to the breakdown of diatomic oxygen, diatomic nitrogen, and water vapor into a mixture of radicals (e.g., atomic oxygen, atomic nitrogen, atomic hydrogen, hydroxyls), metastable species, and ions (e.g., nitrate, nitrite, peroxynitrate). In a cold plasma process, approximately 5% or less of the working gas is ionized. However, cold plasma treatment can generate over 1000 ppm of reactive gas species (RGS). The final result is a range of bactericidal and fungicidal molecules such as ozone, peroxides, nitrates, and many others. This review provides an overview of the mechanisms and chemistry of ACP and its application in inactivating *Aspergillus* spp. and degrading aflatoxins, serving as a novel treatment to enhance the safety and quality of grains and nuts. The final section of the review discusses the commercialization status of ACP treatment.

## 1. Introduction

Grains and nuts such as corn, rice, wheat, oats, barley, peanuts, walnuts, hazelnuts, cashew nuts, etc., are prone to aflatoxin contamination along the agricultural chain from the pre-harvest stage through the harvesting stage and post-harvesting stage. Aflatoxin contamination of these common produces has detrimental consequences on the output of the agricultural sector generally while constraining the attainability of the four food security pillars; availability, access, utilization, and stability [1]. Fungal diseases in crops have a major impact on food security, and there are direct measurable economic consequences [2]. Fungi have the ability to grow on all kinds of grains and nuts, where they cause food spoilage such as toxin production, discoloration, off-flavor development, rotting, and formation of pathogenic and allergenic propagules [3]. Maize farmers in the United States face an annual loss of $160 million due to aflatoxin-related problems. In developing countries, particularly in sub-Saharan Africa, the losses are even greater, totaling $450 million, which accounts for 38% of the global agricultural losses caused by aflatoxin [4].

*Aspergillus* spp. is primarily responsible for aflatoxin production, which can occur at any stage of the crop production chain, influenced by environmental factors such as temperature, humidity, and rainfall, as well as farm management practices like cropping, harvesting, and storage conditions [5]. For example, aflatoxin contamination in maize across China varies by region, with incidence rates ranging from below 40% to over 90% [6]. The average aflatoxin contamination levels range from 1.15 to 107.93 μg/kg. In a separate survey conducted in France, 6% of 114 maize field samples and 15% of 81 maize silo samples tested positive for aflatoxins [7].

Plant stress factors, including drought, injury caused by insects or birds on the host, and inadequate fertilization, are known to accelerate aflatoxin production by *Aspergillus* spp. [8]. The contamination of crops by *Aspergillus* spp. is also influenced by handling and storage conditions at the post-harvest stage [9]. Maize, groundnuts, and tree nuts are common hosts infected by *Aspergillus* spp. at the pre-harvest stage, while wheat, sorghum, and rice are more susceptible at the post-harvest stage [10]. AFB_1_, AFB_2_, AFG_1_, and AFG_2_ are the most common aflatoxins [11]. Agricultural products from Sub-Saharan African countries, Gambia, Uganda, Kenya, and Tanzania and Southeast Asian countries such as China, Thailand, Vietnam, and Indonesia have been associated with the most prevalent occurrence of aflatoxins, which has corresponded to the highest rates (40% for Africa and 27% for Southeast Asia) of hepatocellular carcinoma and aflatoxicosis outbreaks [12]. Aflatoxin contamination in grains and nuts is a worldwide concern due to its toxicological effects on humans and animals [13].

To mitigate aflatoxin contamination, current methods of *Aspergillus* reduction and toxin decontamination in grains and nuts include processes with lengthy treatments, high temperatures, or harsh chemicals. Many of these treatments negatively impact the food causing color loss, texture changes, flavor changes, and lower nutritional content [14]. Aflatoxin is known for its high heat tolerance, remaining stable at temperatures reaching 350 °C; thus, heat-based methods of food processing are insufficient to eliminate the toxicity [15]. Extrusion has degraded 77.6% AFB_1_ from contaminated peanut meals (35.8 ppb); however, the extrusion process at high temperature (150 °C) turned the peanuts into “peanut butter” [16]. A 65.9% reduction in AFB_1_ from the artificially contaminated peanuts (200 ppb) was observed after exposure to a concentration of 6.0 mg/L ozone for 30 min [17]. More than 95% AFB_1_ was reduced from artificially contaminated peanuts (300 ppb) with a combination of microwave and gamma irradiation; however, the process impacted peanut quality such as moisture content, color, hardness, and peroxide value [18].

Atmospheric cold plasma (ACP) is an emerging non-thermal food processing technology, which can deliver beneficial results in a short treatment time and with minimal inputs of air and electricity, without heating effects on the quality of food products, and leaves no residues compared to conventional decontamination technologies [19]. The application of ACP can inactivate *A. flavus* and reduce aflatoxin from contaminated commodities, such as pistachio [20,21,22], hazelnuts [23,24,25,26,27], and maize [28]. A hazelnut study has shown that *A. flavus* and *A. parasiticus* were reduced by 4.50 log CFU/g and 4.19 log CFU/g, respectively, after 5-minute treatments at 655 Watts with dry air as the plasma-forming gas [23].

In this review, cold plasma technology for the inactivation of *Aspergillus* spp. and degradation of aflatoxins and their mechanism of decontamination have been elaborated. This review has also focused on the application of cold plasma in grains and nuts. Furthermore, the commercialization of ACP treatment has been discussed.

## 2. Atmospheric Cold Plasma

In recent years, consumers have increased demands for convenient, nutritious, safe, and long shelf-life foods. Existing traditional thermal food processing technologies make it difficult to meet these requirements; therefore, innovative non-thermal food processing and preservation technologies are expected [29]. ACP as a simple, fast, and cost-effective technology with minimized food quality impacts, has been drawing more attention. Current cold plasma research for the food industry focuses on its applications for food decontamination and contaminant degradation [30,31].

Plasma is an ionized gas, generated by adding energy via light, heat, or electricity to a gas. During the process of plasma generation, electrons are stripped away from their respective atoms resulting in ionization. As more energy is added, more “plasma” is generated. Plasma is defined as the fourth fundamental state of matter. When the gas is fully ionized, the created plasma contains light, photons, ions, electrons, free radicals, reactive species, and electrically charged particles [32]. ACP (also known as non-thermal or low-temperature plasmas) are partially ionized gas where approximately 5% or less of the gas exists as plasma [33]. Depending on the system applied, the mechanisms of cold plasma may involve a combination of electric field, ultraviolet light, and reactive gas species. The RGS generated may include ions (H^+^, H_3_O^+^, O^+^, H^−^, O^−^, OH^−^, N_2_^+^, etc.), molecular species (N_2_, O_2_, O_3_, H_2_O_2_, etc.), and reactive radicals (O•, H•, OH•, NO•, etc.) [21]. RGS with varying stability and half-lives are produced by ACP. Table 1 provides examples of some reactive species along with their half-lives. Species with a half-life of less than 0.5 s are classified as short half-life species [34]. ACP can be generated in air and other gases at room temperature and atmospheric pressure with low energy consumption.

### 2.1. Dielectric Barrier Discharge (DBD)

One of the most popular techniques for producing ACP is DBD. Here, a dielectric material is applied to one or both electrodes arranged in a parallel plate configuration to create a plasma discharge between them (Figure 1a). In DBD-ACP, the current discharge (i.e., electrical arcs) is limited by the dielectric barrier layer, such as alumina, glass, silica glass, ceramic materials, and thin enamel or polymer layers, which cover one or both electrodes, or it can be suspended between two electrodes; hence, a spark or arc discharge can be avoided. This allows the electric field to interact with the working gas while preventing excess heating from sparks or arcs. The RGS generated is dependent on the equipment setup and process parameters employed such as voltage, working gas, electrode shape, electrode gap, gas volume, food type, packaging materials, and the exposure mode (direct or indirect). The versatility of geometrical configurations and miniaturization, operational parameters, simple design, cost, safety, and characteristics of the power supply are only a few of the many benefits that DBD offers [41]. Dielectric barriers serve as a stabilizing material, preventing arc transitions and aiding in the production of a large number of micro-discharges for homogenous treatments [41]. A DBD-ACP treatment was applied to pistachios contaminated with *A. flavus*. The treatment utilized air as the working gas, with an electric power consumption of 2.49 W/cm^3^ and a voltage of 10.7 kV. The pistachios were exposed to the plasma for 18 min, resulting in *A. flavus* from the samples being reduced to an undetectable level. However, no quality evaluations were conducted to assess the impact of the treatment on the pistachios’ sensory or nutritional properties [22]. Another study used the DBD-ACP treatment on corn kernels to remove AFB_1_ from the contaminated surface. The ACP was operated at two different discharge power levels: low (0.18 W/cm) and high (0.31 W/cm). Treatment durations varied, including 30, 60, 120, 240, and 480 s, with air as the working gas. The treatment successfully achieved 100% removal of AFB_1_ from the corn kernels [42]. Therefore, the DBD system is simple and could be easily scaled up for high production volume.

### 2.2. Atmospheric Pressure Plasma Jets

The plasma jet can realize the operation of gas discharge in a non-sealed electrode arrangement and the projection of the discharge plasma species into an open environment [45]. The distinctive feature of the atmospheric pressure plasma jet configuration is its ability to launch stable plasma species into a separate environment, where the electric field can be very low [46]. This system usually consists of two concentric rings or coaxial electrodes with a gap of 0.5–6 mm, in which the gas flows between the pair of electrodes at a range of several liters per minute [45]. The outer electrode is grounded, and the central electrode is excited by radiofrequency, typically 13.56 MHz, accelerating the free electrons that collide with gas molecules forming various reactive species [47]. The gas flowing at a high flow rate, usually a noble gas or a mixture with a reactive gas, pushes the plasma formed outside the electrode region, projected as a jet, and discharges plasma species into the open environment [48]. The plasma jet produces a stable, homogenous, and uniform discharge at atmospheric pressure. A plasma jet treatment was applied to 5 g of hazelnuts contaminated with 3 µg/kg of AFB_1_. The treatment was operated with dry air (purity 99.99%) at 3000 L/h of gas flow rate, 25 kHz frequency, 655 W of power, and a 7 cm distance between the plasma jet and the sample. The hazelnuts were treated for 1.7 min at room temperature, resulting in a 72% reduction in AFB_1_ levels. However, there were no data provided on the plasma jet or hazelnut surface temperature and whether some or all of the reported reductions came from thermal effects. Importantly, the treatment did not compromise the sensory attributes of the hazelnuts, as assessed by a sensory panel, indicating that the plasma jet technology can effectively reduce harmful contaminants like AFB_1_ without negatively impacting the food’s quality or taste [25,26,49]. Another study was performed using atmospheric pressure plasma jet treatment on 10 g of maize contaminated with *A. flavus* and *A. parasiticus* [50]. The plasma reactor was made from glass with a 49 mm diameter, 147 mm length, and an L/D ratio of 3. A stainless steel stand was designed to hold the reactor on top of the plasma jet (a nozzle with an inner diameter of 4 mm). The generated plasma expanded to the outside of the nozzle with a length of 20 mm. The treatment used dry air as the working gas, with a flow rate of 3000 L/h, a voltage of 10 kV, and a power output of 655 Watts at room temperature. The maize was exposed to the plasma jet for 5 min, resulting in a significant reduction in microbial contamination. Specifically, *A. flavus* was reduced by 5.48 log CFU/g, while *A. parasiticus* showed a 5.20 log CFU/g reduction. However, there were no data provided on the surface temperature and whether some or all of the reductions came from thermal effects. The quality evaluations were conducted to assess the impact of the treatment on the maize’s sensory or nutritional properties [23,50]. However, this plasma is limited when aiming at a large area, but the arrangement of several jets must be applied in this case.

### 2.3. Gliding Arc Discharge

Generating a gliding arc discharge requires a high-velocity gas flowing between two or more divergent electrodes (electrode gap of 1–5 mm) that are connected to a high-voltage power supply (1 kV to 10 kV or higher, and 1–10 kHz frequency) [51]. The gas, such as humid air, pushes the formed arc plasma in the same direction along the axis of the reactor resulting in a series of sliding discharges that stretch across the inter-electrode region. When the gas flow rate between the electrodes increases, it extends the length of the arc along the electrodes and requires larger power to maintain the arc. The plasma gas temperature generated from the gliding arc is several hundred degrees Celsius [52]. During gliding arc discharge plasma, a variety of reactive species are generated due to the interaction of the electrical discharge with the working gas, which is typically air or a mixture of gases. Key reactive oxygen species (ROS) produced include ozone (O_3_), singlet oxygen (^1^O_2_), and oxygen atoms (O), all of which arise from the dissociation of oxygen molecules and subsequent recombination or energy transfer [53]. Additionally, peroxy radicals such as HO_2_ and O_2_• are formed, further enhancing the reactivity. In terms of reactive nitrogen species (RNS), nitric oxide (NO) and nitrogen dioxide (NO_2_) are generated through reactions between nitrogen and oxygen in the plasma, while nitrous oxide (N_2_O) and peroxynitrite (ONOO^−^) can also form under certain conditions. Along with these, highly reactive radicals like hydroxyl radicals (OH•), superoxide radicals (O_2_•^−^), and hydrogen peroxide (H_2_O_2_) are produced, particularly when water vapor is present in the plasma. Excited species, including excited oxygen (O_2_*) and nitrogen molecules (N_2_*), also play a role in the generation of these reactive species [51,53]. These species contribute to the antifungal effects of gliding arc plasma, as they can damage microbial cell membranes, proteins, and nucleic acids, leading to the inhibition or destruction of *Aspergillus* spp. A gliding arc cold plasma system consists of three parts: the plasma reactor, the mixing of dry air with wet air, and the voltage measurement and recording system, which was used to inactivate *A. flavus* and *A. parasiticus on the wheat* [53]. The plasma reactor includes two stainless steel tubular electrodes with an external diameter of 6 mm and a radius of curvature of 180 mm. These electrodes were positioned at an upper distance of 2 mm and a lower distance of 80 mm from each other, enclosed within a Teflon and glass frame. The reactor bed, where wheat samples were situated, was at a distance of 70 mm from the discharge opening. The treatment was conducted at a frequency of 20 kHz and using dry air (21% oxygen and 79% nitrogen) as the working gas, with an added humidity of 1000 ppm water vapor concentration inside the reactor. The reactor was stated to operate at room temperature; however, there were no data provided on the plasma jet or wheat surface temperature and whether some or all of the reported reductions came from thermal effects. The plasma treatment resulted in significant reductions in fungal growth, with *A. flavus* showing a 78.74% reduction and *A. parasiticus* a 68.57% reduction. Notably, the plasma treatment inhibited fungal growth by extending the lag phase and reducing the overall growth rate of the fungi, highlighting its potential as a viable alternative to fungal contamination [53]. However, the lower the gas flow rate, the greater the NOx concentration produced in the water [54].

### 2.4. Microwave Discharge Plasma

Microwave discharges are the electrical discharges generated by electromagnetic waves with frequencies exceeding 300 MHz. In microwave plasma generators, electromagnetic waves typically at 2.45 GHz are emitted by a magnetron and used to produce microwave discharges, with a power of 500 W to 2 kW, a voltage (Magnetron) of 2 kV to 4 kV, a gas temperature of 300 K to 1000 K for non-thermal plasma, a gas flow rate of 1–10 L/min, and a pressure of 0.1 Torr to atmospheric pressure [44,55]. The microwave electric field accelerates the electrons of gas molecules and forms cold plasma as a result, without electrodes. This system can produce plasma at low atmospheric pressures. Some important advantages of using microwaves for cold plasma production include increased electron density, high efficiency in generating reactive species, and contamination-free processes [56]. At the plasma–water interphase, surface-wave microwaves generate plasma discharge. Varying the initial gas mixture composition (Ar, Ar/N_2_/O_2_ binary, and ternary mixture) and the distance between the edge of the plasma-discharge tube and the water surface altered the electron and species densities at the interaction between the plasma and water. H_2_O_2_ was undetected in plasma-activated water (PAW) via the microwave plasma system; however, ONOO^−^ was generated as an isomer of NO^3−^. A study was conducted using the afterglow of a low-pressure Ar/N_2_/O_2_ (N_2_-20%O_2_, N_2_-10%O_2,_ N_2_-2%O_2,_ Ar-20%O_2_, Ar-40%O_2_, and Ar-20%O_2_+N_2_-2%O_2_) microwave discharge at 200–800 Pa pressure, rich in oxygen and nitrogen atoms, O_2_ and NO molecules, and UV radiation. The study explored the impact of ROS and RNS on barley and corn seed germination, as well as their surface disinfection from two types of *Fusaria*, *Fusarium graminearum*, and *Fusarium verticillioides*. The results showed that the germination and vigor of non-infected seeds were not significantly affected when barley was treated for 120 s at 200 Pa (only 2% of the pressure of one atmosphere) and corn for 240 s at 400 Pa. Afterglow treatments, independently of the mixture composition, do not significantly influence the germination and the vigor of seeds, although they show a slightly decreasing trend for longer treatment times [56]. However, the seeds were effectively decontaminated from the germination inhibitors, *Fusarium graminearum,* and *Fusarium verticillioides*, when barley was treated for 3 min at 400 Pa with an 80% Ar-20%O_2_ discharge afterglow, and corn for 4 min at 800 Pa in the same gas mixture, followed by a 2-min treatment with an N_2_-2%O_2_ afterglow. The temperature of the gas was lower than 40 °C [56]. These treatments also improved the germination of the infected seeds from 35% and 50% to above 80%, respectively. Additionally, it was found that high NO content in the mixtures and surface heating of the seeds due to the recombination of O and N atoms inhibited barley germination. These findings highlight the importance of carefully optimizing plasma species to effectively decontaminate.

## 3. Application of ACP Against *Aspergillus flavus* and Aflatoxins in Nuts and Grains

### 3.1. Nuts

In recent years, there has been increased interest in the potential applications of ACP for microbial decontamination and mycotoxin treatments, including *A. flavus* inactivation and aflatoxin degradation from contaminated commodities, such as pistachios [20,21], hazelnuts [24,27], and peanuts [57]. Promising results of *Aspergillus* inactivation and aflatoxin degradation have been reported. Table 2 summarizes the recent research using cold plasma to inactivate *Aspergillus* spp. in nuts. The plasma jet has successfully reduced 5.14 log CFU/g from pistachios with a 15 min treatment [20], and 4.64 log CFU/g from hazelnuts after a 1.7 min treatment [25]. The temperature of the treated samples was measured at 20 kV due to the higher temperature of the plasma jet at this voltage and recorded at 27 °C. The quality properties of pistachios such as moisture content, peroxide value, free fatty acid, and total polyphenol were measured after plasma treatment and compared with untreated samples, which indicated no significant differences. Meanwhile, a sensory evaluation was performed to determine the acceptance of the color, smell, and taste of plasma-treated pistachio samples, which received satisfactory scores [21]. No quality experiments were conducted; instead, they performed a sensory evaluation and reported satisfactory scores for texture, overall color, odor, and overall acceptance of plasma-treated hazelnuts. A plasma jet was also applied to inactivate both *A. flavus* and *A. parasiticus* from hazelnuts [23] with the same treatment conditions. After 655 W and 5 min dry-air plasma treatments, 4.50 log CFU/g of *A. flavus* and 4.19 log CFU/g of *A. parasiticus* were decontaminated from hazelnuts. However, quality parameters of plasma-treated hazelnuts were not conducted.

DBD was widely applied to treat different food commodities. A study reported that 4.0 log CFU/g *A. flavus* from pistachio was reduced using DBD with 130 W and 3 min [21]. However, after 3-minute plasma treatment, the instrumental color value a* (redness) increased considerably and b* (yellowness) decreased slightly, and the overall color changes resulted in a darker pistachio nuts color. But the antioxidant activity of pistachio nuts slightly increased after treatment. Another research conducted by researchers [21] also applied DBD to inactivate *A. flavus* spores from artificially contaminated pistachio surfaces and indicated *A. flavus* was completely inactivated after an 18 min treatment. Meanwhile, the temperature, pH, moisture content, and color of pistachio samples remained unchanged after plasma treatment. Moreover, we applied DBD to inactivate *A. flavus* from raw hazelnuts using different working gases [24]. After 400 W and 3 min treatments, 2.95 log CFU/g and 2.56 log CFU/g were reduced using ambient air (21% O_2_ + 78% N_2_) and synthetic air (20%O_2_ + 80%N_2_) without water vapor, respectively. Contrarily, *A. flavus* spores were reduced to below the detection limit (1.0 log CFU/g) from an initial population of 4.6 CFU/g when using synthetic air with water vapor to generate plasma. Although their treatments effectively inactivated *A. flavus* from hazelnuts, there were no quality experiments conducted after plasma treatments (Table 2). A treatment of 5 min resulted in a 2.20 log spores/sample reduction of *A. flavus* spores on raw peanut kernels. *A. flavus* was almost completely inactivated (99.9%) by DBD treatment for 10 min with 80% RH in the air and 24 h post-treatment storage without adversely affecting peanut quality [57]. Scanning electron microscopy (SEM) was applied to investigate the impact of atmospheric pressure plasma treatment on the morphology of *A. flavus* [20,25]. They all concluded that active plasma species disintegrated the cell wall and cell contents such as cytoplasmic structures were released from plasma-treated *A. flavus* spores. Cold plasma’s antimicrobial effect is a result of the action of charged particles and reactive species generated in the plasma, which can cause damage to the cell membrane, leading to further penetration of reactive species into the cell, DNA damage, and the disruption of chemical bonds [29]. In a light microscopic analysis of *A. flavus* cells treated with a radiofrequency plasma jet, the Study reported that the conidiophores and vesicles were observed to be broken after plasma treatment, resulting in cell leakage and loss of viability [50].

Table 3 illustrates the recent work with cold plasma on the degradation of aflatoxins in nuts. A study explored the AFB_1_ degradation effect of a plasma jet on pistachio kernels [26]. The authors observed that up to 80% AFB_1_ was reduced after 15 min of atmospheric plasma treatment. Pistachio quality and sensory evaluation were also assessed after plasma treatment and indicated no significant differences between plasma-treated samples and controls. In a study, a plasma jet was applied to 5 g of hazelnuts containing 3 µg/kg of AFB_1_ [26]. The treatment parameters included a power of 655 W, air as the working gas, a flow rate of 3000 L/h, and a treatment duration of 1.7 min. The results showed a 72% reduction in AFB_1_ levels. Additionally, the sensory attributes of the hazelnuts were maintained, as evaluated by a sensory panel, indicating that the plasma treatment did not negatively affect the quality of the food. A DBD system was applied with N_2_ to generate plasma to degrade AFB_1_ from hazelnuts [27] resulting in total aflatoxins and AFB_1_ reductions of over 70% after 12 min of treatment. The authors reported that AFB_1_ and AFG_1_ were more sensitive to plasma treatments compared to AFB_2_ and AFG_2_, respectively. AFB_1_ was more sensitive to plasma treatments compared to AFG_1_. However, there was no quality analysis of plasma-treated hazelnuts. A 71.3% AFB_1_ reduction in the raw peanut kernels was achieved with a DBD treatment for 2 min and 80% humid air as the working gas without post-treatment. The reduction of AFB_1_ toxin also significantly increased with increasing treatment time, higher RH and post-treatment storage. There were no statistical differences in the color, texture, and peroxide value between treated and untreated peanuts [57]. A DBD system was used to treat AFB_1_ contaminated pistachios. According to the study, the maximum AFB_1_ reduction rate (52.42%) was observed after 130 W and 180 s [21]. After applying ACP to treat pure AFB_1_ in 40% RH air for 5 min, six main degradation products were identified (C_16_H_17_O_6_, C_17_H_15_O_7,_ C_14_H_13_O_5_, C_14_H_11_O_6_, C_17_H_13_O_7,_ C_19_H_19_O_8_) and it was stated that AFB_1_ was degraded by cold plasma by targeting the furan ring, lactone ring, cyclopentanone, and the methoxyl group [57,61].

### 3.2. Grains

Atmospheric cold plasma treatment has been proven to decontaminate aflatoxigenic fungi (*A. flavus and A. parasiticus*) on corn [50,63,64], rice [65], and wheat [53,63]. Table 4 summarizes the recent research using cold plasma to inactivate *Aspergillus* spp. in grains. A 10 min treatment at 85 kV reduced the number of viable *A. flavus* spores by more than 95% [15]. The optical emission and absorption spectroscopy demonstrate ionization of hydroxyl groups, atomic oxygen and nitrogen, and confirm the production of ROS and RNS, e.g., O_3_, NO_2_, NO_3_, N_2_O_4_, and N_2_O_5_. Treated *A. flavus* spores show surface ablation and membrane degradation by scanning electron microscopy. An atmospheric pressure fluidized bed plasma (APFBP) system was used for the decontamination of maize artificially contaminated with *A. flavus* and *A. parasiticus* spores [50]. After treatment of 5 min and using air as the working gas, the maximum reduction was 5.48 and 5.20 log CFU/g for *A. flavus* and *A. parasiticus,* respectively. The temperature of maize after treatment was below 45 °C, which could be considered as a suitable value for “non-thermal” treatment of food materials. But the quality of treated maize samples was not assessed. Another study investigated the use of gliding arc plasma to control the growth of *A. flavus* and *A. parasiticus* on wheat [53]. The plasma treatment was performed using dry air as the working gas, consisting of 21% oxygen and 79% nitrogen, with the injected air containing 1000 ppm of water vapor for humidity inside the reactor. The results showed that plasma treatment led to a significant reduction in fungal growth, with a 78.74% decrease in *A. flavus* and a 68.57% decrease in *A. parasiticus*, respectively. The plasma treatment exhibited its fungal inhibitory effect by extending the lag phase and reducing the growth rate of the fungi. But the temperature and quality of treated wheat have not been evaluated in this study.

A low-pressure cold plasma (LPCP) system was applied to decontaminate *Aspergillus* spp. in wheat, oat, barley, and corn using air and SF_6_ as working gases [63]. The running pressure was 66.7 Pa and the applied power was around 300 Watts. A reduction of 3.0 log CFU/g of *Aspergillus* spp. was achieved within a treatment time of 15 min while preserving the germination quality of the seed. The temperature of maize and gas plasma after treatment was not tested. Although the physiological or biochemical basis for inactivation is unknown, air or SF_6_ plasma is biocompatible which means that damage to living seeds is avoided and offers a suitable treatment for the surface disinfection of seeds [63]. A 20-min DBD treatment with pure O_2_-fed, voltage of 13 kV and frequency of 3 kHz resulted in a reduction of conidial germination and subsequent growth of *A. flavus* on maize by 50–90% [64]. However, the effect of plasma on the temperature and quality of maize has not been evaluated in the study.

Aflatoxin contamination of cereal grains is a global food safety issue. Cold plasma degradation of mycotoxins is a promising approach [66]. The aflatoxin degradation pathways using cold plasma treatment are inevitably related to their molecular structure, the nature of the plasma chemistry, and thus, the species interaction with toxin molecules [31]. Aflatoxin destruction using cold plasma has focused on multiple aspects, including plasma source, working gas, sample type, etc. A summary of research studies demonstrating aflatoxin degradation in grains using cold plasma is provided in Table 5. The reactive gas species such as ozone, hydroxyl, and free radicals (O•, H•, OH•, NO•, etc.) produced by plasma treatment contribute to aflatoxin degradation [28]. AFB_1_ is one of the main mycotoxins that are usually found in cereals [67]. Artificially spiked AFB_1_ was effectively degraded from corn by using DBD treatment, and it was reported that more than 62% AFB_1_ was degraded by 1 min treatment in 40% RH air. In another study, a corona discharge plasma jet was used to degrade AFB_1_-spiked food commodities for 30 min and reducing 56.6% and 45.7% from wheat and rice with the initial toxin concentration levels of 7.14 µg/kg and 5.97 µg/kg of AFB_1_, respectively [49]. The working gas used was air, with an air blower rotational speed of 3312 rpm, a treatment time of 30 min, and a voltage of 20 kV. The oxidation involving free radicals in plasmas may have played a dominant role in AFB_1_ degradation. However, no quality evaluations and temperature measurements were conducted to assess the impact of the treatment on the rice and wheat.

A high-voltage (80 kV) ACP treatment on degradation of aflatoxin in corn (Table 5). Gas type (Air, MA65), relative humidity (5, 40, 80% RH), treatment time (1, 2, 5, 10, 20, and 30 min), mode of reaction, and post-treatment storage were applied and results showed that aflatoxin in corn was degraded by 62% and 82% by 1- and 10-min ACP treatment in 40% RH air, respectively [28]. Higher degradation of aflatoxin was achieved by the treatment with MA65 as a working gas, at higher relative humidities (40% and 80%). The temperature of filled gases changed from 22 °C to around 38 °C. To determine the effect of DBD cold plasma on AFB_1_ and the physicochemical characteristics of oats, whole oat grains were subjected to plasma treatment for 1, 3, and 6 min at powers of 32, 37, and 42 W, respectively [68]. Untreated samples were also evaluated as controls. The results showed that DBD treatment had the greatest reduction in 6 min at a power of 37 W which reduced AFB_1_ concentration by 82%. Furthermore, there was no significant difference in some qualitative parameters such as protein and pH compared with the control sample, whereas the peroxide value and β-glucan increased and alpha-tocopherol decreased [68]. Another report on the complete degradation of AFB_1_ from artificially contaminated corn kernels involved an indirect DBD treatment working in ambient air. Surface analysis of the corn kernels revealed the presence of minor changes in the treated samples, indicating that indirect DBD treatments induced slight oxidation of the corn kernel surface [42].

**Table 4 toxins-17-00129-t004:** Summary of literature on cold plasma for inactivation of *Aspergillus* spp. in grains.

Plasma Source	Sample Characteristics	Organism(s)	Treatment Conditions	Significant Findings	References
Fluidized bed plasma	10 g maize	*A. flavus* and *A. parasiticus*	Working gas: air and N2 Flow rate: 3000 L/h Voltage: 10 kV Power: 655 W Treatment time: 5 min	Reduction of 5.48 log CFU/g for *A. flavus* and 5.20 log CFU/g for A. parasiticus, respectively Temperature inside the fluidized bed reactors during the plasma process below 45 °C No quality evaluations conducted	[50]
Plasma jet	Brown rice bar	*A. flavus*	Working gas: 60% argon -40% air Voltage: 20 kV Treatment time: 15 min Flow rate: 3 L/min gas Power of 20 W and 40 W Treatment time 5, 15 and 25 min	Plasma power of 40 W and exposure time of 20 min required to reduce mold growth of approximately 4.0 log CFU/g Neither the temperature of the plasma gas nor the treated brown rice bar was reported	[61]
DBD	Maize seeds	*A. flavus*	Working gas: pure O^2^ (air-liquid, 99.999%) and synthetic air (air-liquid, 99.999%) Voltage: 13 kV Frequency: 6 kHz Treatment time: 3 and 20 min	50–90% reduction of conidial germination and subsequent growth of *A. flavus* on maize by air-fed plasmas Neither the temperature of the plasma gas nor the treated groundnuts was reported No quality assessment	[64]
Gliding arc plasma	Wheat	*A. flavus* and *A. parasiticus*	Working gas: dry air, consisting of 21% oxygen and 79% nitrogen. Humidity of the injected air (1000 ppm water vapor concentration) inside the reactor	78.74% and 68.57% reduction in the growth of *A. flavus* and A. parasiticus, respectively Plasma induced a fungal inhibitory effect by prolonging the lag time and decreasing the growth rate Neither the temperature of the plasma gas or the treated wheat was reported	[53]
Very Low pressure cold plasma (LPCP)	Wheat, oat, barley, corn	*A. flavus*	Working gas: air, SF6 Pressure: 66.7 Pa (0.0006 atm) Frequency: 1 kHz Voltage: 20 kV Power: 300 W Treatment time: 5, 10, and 20 min	3.0 log CFU/g reduction achieved within 15 min of treatment Seed germination retained after treatment No temperature test after treatment	[63]

**Table 5 toxins-17-00129-t005:** Application of cold plasma for aflatoxin degradation in grains.

Plasma Source	Sample Characteristics	Treatment Conditions	Detection Technique	Significant Findings	References
Plasma jet	50 g rice (5.97 µg/kg AFB_1_)	Working gas: air Air blower rotational speed: 3312 rpm Voltage: 20 kV Frequency: 58 kHz Treatment time: 30 min	HPLC	45.7% AFB1 reduction in rice No quality evaluations conducted Neither the temperature of the plasma jet nor the treated rice was reported.	[49]
Plasma jet	50 g wheat (7.14 µg/kg AFB_1_)	Working gas: air Air blower rotational speed: 3312 rpm Voltage: 20 kV Frequency: 58 kHz Treatment time: 30 min	HPLC	56.6% AFB1 reduction in wheat No quality evaluations conducted Neither the temperature of the plasma gas nor the treated wheat was reported	[49]
Gliding arc plasma	Wheat	Working gas: dry air, consisting of 21% oxygen and 79% nitrogen Treatment time: 12 min Power: 10.88 W	HPLC	64%, 41%, 59%, 40%, and 61% reduction for AFB1, AFB2, AFG1, AFG2, and total aflatoxin, respectively Neither the temperature of the plasma gas nor the treated wheat was reported	[53]
DBD	Artificially spiked 100 uL AFB_1_ solution in chloroform (50 μg/mL) on 25 g corn kernels	Working gas: air, and MA65 Voltage: 90 kV Relative humidity: 5, 40, 80% RH Treatment time: 1, 2, 5, 10, 20, and 30 min Post-treatment storage: 0 and 24 h	Lateral Flow Devices (LFDs)	AFB1 degraded by 62% and 82% with 1 and 10 min in 40% RH air and 24 h post-treatment, respectively Higher AFB1 degradation achieved in MA65 at higher relative humidities The maximum temperature of plasmas was 38 °C No quality evaluations conducted	[28]
DBD	20 g rice (AFB_1_, AFB_2_, AFG_1_, AFG_2_ of 20 μg/kg, 50 μg/kg, 100 μg/kg, 150 μg/kg)	Working gas: 0%O_2_ + 70%N_2_ + 30%CO_2_, 25%O_2_ + 45%N_2_ + 30%CO_2_, 45%O_2_ + 25%N_2_ + 30%CO_2_, and 65%O_2_ + 5%N_2_ + 30%CO_2_ Voltage: 70–100 kV Treatment time: 5–60 min Input power: 220 V-50 Hz Relative humidity: 56 ± 1%	UPLC H-Class/Xevo TQ-S	Reduced 55.34% of AFB1 and 56.37% of aflatoxins in rice Degradation rate of aflatoxins increased with the increase in oxygen content Neither the temperature of the plasma gas nor the treated rice was reported No significant impact on rice quality except for fat oxidation	[69]
DBD	Oat	Working gas: Argon Gas flow rate: 1 NL/min Voltage: 1–10 kV Treatment time: 1, 3, and 6 min Power: 32, 37, and 42 W	HPLC	Treatment for 6 min at a power of 37 W reduced AFB1 concentration by 82% Increased the amount of peroxide and β-glucan, decreased alpha-tocopherol and viscosity of oat flour Improved the farinograph properties of oat flour Neither the temperature of the plasma gas nor the treated oat was reported	[68]
DBD	Corn kernels inoculated with 20 µL AFB_1_ (33 µg/mL)	Working gas: air Powder: 0.18 and 0.31 W/cm Treatments time: 30, 60, 120, 240, 480 s	Not available	100% AFB1 removed from the contaminated corn kernels No significant chemical or morphological modifications of the corn kernel surface Neither the temperature of the plasma gas nor the treated corn was reported.	[42]
DBD	Maize seeds spiked with 0.25 mL of AFB_1_ solution (1 μg/mL)	Working gas: pure O_2_ (air-liquid, 99.999%) and synthetic air (air-liquid, 99.999%) Voltage: 13 kV Frequency: 6 kHz Treatment time: 3 and 20 min	HPLC-HRMS	AFB1 degradation of 23% on kernels achieved after 20 min with pure O_2_ process Neither the temperature of the plasma gas nor the treated maize was reported. No quality assessment of treated maize	[64]

## 4. Commercialization of ACP Technology

Cold Plasma technology has a current global market value of 3.2 billion dollars with a Compound Annual Growth Rate (CAGR) of 18% [70]. Much of this market value is in semiconductor manufacturing with growing applications in medicine such as skin rejuvenation and wound healing. The application of cold plasma for the treatment of grains, nuts, fruits, vegetables, and seeds remains a very small global market of approximately 5%, but agricultural applications are growing. Currently, there are several seed treatments utilizing cold plasma including ActivatedAir^TM^ (Zayndu, Ltd., Loughborough, UK) and Clean Crop Technologies (Holyoke, MA, USA).

Commercial cold plasma technology available for mold control and mycotoxins remains limited. For example, NanoGuard Technologies (Louis, MO, USA) claims their cold plasma technology can significantly reduce mold on grains and seeds and reduce aflatoxin by 50% or more [71]. There is no information available on their website regarding whether their technology has been deployed in the grain industry. The Cold Plasma Group (Kingston, ON, Canada) advertises a cold plasma treatment chamber available for use in reducing molds and bacteria on cannabis and cannabis products without reducing the volatile compounds [72]. But their website lacks details on commercialization installation.

In the next five years, exponential growth in cold plasma applications is expected in food safety, food preservation, and food production because of the versatility of the technology to transform air into plant fertilizer, eliminate airborne and surface contaminants, including toxins and molds, while minimally impacting the nutritional value. There are three primary barriers to industry adoption:

(1) The lack of commercial-scale cold plasma equipment available for purchase. Hundreds of research laboratories have developed cold plasma technologies, but food manufacturers want to purchase cold plasma equipment proven to work for their applications. Designing these commercial-scale systems requires manufacturing partners willing to invest in tooling and specialized high-voltage generators with a future return on investment. Existing cold plasma technologies used in medical and semiconductor manufacturing are not suitable for large food processing facilities.

(2) The sequence of steps needed to demonstrate the safety of cold plasma-treated food is highly variable. Currently, regulations under the FDA, USDA, and EPA may apply to cold plasma technologies. Agricultural and food companies wanting to introduce this new technology into their manufacturing process, have to navigate these uncharted waters. There is significant risk, as with any new technology, in generating adequate data and compelling data to demonstrate a ‘de minimis’ effect on the food. The regulations are complex beyond the scope of this review, but suffice it to say that they must be understood and followed to ensure a safe food product.

(3) The selectivity of the cold plasma process is very product- and process-dependent. Commercial adoption of cold plasma technology requires a complete understanding of the reactive gas species generated based on the plasma source (gas, voltage, electrodes, treatment time, etc.); the interaction of the reactive gas species with the food; and the reactive gas species reversion process after treatment. For example, what happens with those reactive gas species remaining which do not interact with the food? Are they exhausted on the outside, recaptured, and reused? Are they released inside the facility potentially exposing workers? There are many complex chemistries of gas, solid, and liquids that are needed in the design of a cold plasma treatment for food. Also, the cold plasma process needs to be robust, such that it can guarantee effectiveness regardless of the variability in the incoming grain or seeds.

As more startups and established companies expand into cold plasma equipment design and agricultural applications, it is expected that a cold plasma device manufacturing sector will be established, and as the technology matures, equipment costs will come down and equipment reliability will increase.

## 5. Conclusions

*Aspergillus* spp. are commonly found in agricultural products such as grains and nuts, producing aflatoxins, which are harmful mycotoxins. Aflatoxins, particularly B_1_, B_2_, G_1_, and G_2_, pose serious health risks to both humans and animals. ACP is gaining attention as a non-thermal, effective method for decontaminating food products. ACP is created by applying an electrical charge to a gas (like air), transforming it into a reactive gaseous state, and generating substances like ozone (O_3_), hydroxyl radicals (OH), and nitrogen oxides. These reactive species are highly effective in eliminating *Aspergillus* spp. and breaking down aflatoxins, presenting ACP as a powerful alternative to traditional methods. ACP treatment can significantly reduce aflatoxin contamination, making it a viable alternative to traditional chemical or thermal treatments. ACP offers a safe, environmentally friendly way to meet food safety standards without compromising the quality of the food, making it a promising tool in the fight against mycotoxin contamination. The commercialization of cold plasma treatment is expected to grow in the coming years as the technology matures and more industries begin to recognize its potential benefits. Moreover, further research needs to be conducted on cold plasma to establish process guidance for a number of food products, and understand the chemistry of plasma and its interaction with microbes and food matrices which are essential in optimizing the process for individual applications.

## Figures and Tables

**Figure 1 toxins-17-00129-f001:**
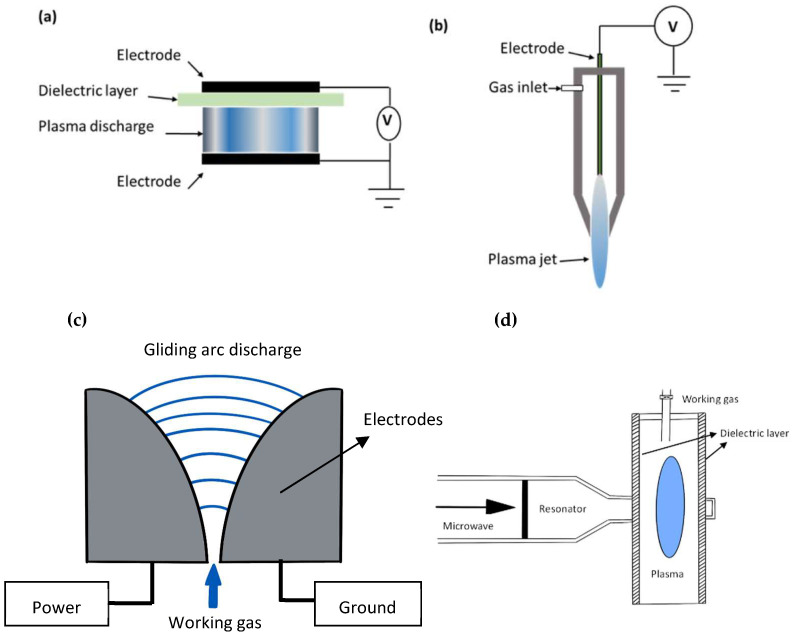
Schematic diagram of dielectric barrier cold plasma (**a**), plasma jet system (**b**) [31], gliding arc discharge (**c**) [43], and microwave discharge (**d**) [44].

**Table 1 toxins-17-00129-t001:** Examples of reactive oxygen and nitrogen species produced during ACP treatment and their half-life.

Reactive Species	Half-Life	References
ONOO^•−^ (peroxynitrite)	5–20 milliseconds	[34]
O_2_^•−^ (superoxide)	~1 microseconds	[34]
H_2_O_2_ (hydrogen peroxide)	seconds to minutes	[34]
O_3_ (ozone)	99 min under refrigeration	[35]
NO_3_ (nitrate)	5–8 h (human blood)	[36]
NO_2_ (nitrogen dioxide)	13 min (when O_3_ < 1.0 ppm)	[37]
NO (nitric oxide)	15 s	[38]
N_2_ (C^3^π_u_)	38 nanoseconds	[39]
N_2_ (B^3^π_g_)	13 nanoseconds	[39]
OH• (hydroxyl radical)	~10^−9^ s	[40]

**Table 2 toxins-17-00129-t002:** Summary of literature on cold plasma for inactivation of *Aspergillus* spp. in nuts.

Plasma Source	Sample Characteristics	Organism(s)	Treatment Conditions	Significant Findings	References
Plasma jet	10 pieces pistachio kernels	*A. flavus*	Working gas: 62% argon/38% air Voltage: 20 kV Treatment time: 15 min Flow rate: 3 L/min gas	~5.14 log CFU/g reduction No effect on quality characteristics of pistachio kernels The temperature of treated pistachio was 27 °C	[20]
Plasma jet	10 g hazelnuts	*A. flavus* and *A. parasiticus*	Working gas: air Flow rate: 3000 L/h Power: 655 W Frequency: 25 kHz Treatment time: 1.7 min	5.4 and 5.5 log CFU/g of reductions in *A. flavus* and *A. parasiticus*, respectively No quality evaluations conducted Neither the temperature of the plasma jet or the treated hazelnuts was reported.	[23]
Plasma jet	10 g hazelnuts	*A. flavus* and *A. parasiticus*	Working gas: dry air Flow rate: 3000 L/h Power: 655 W Frequency: 25 kHz Treatment time: 5 min	4.50 and 4.19 log CFU/g reduction for *A. flavus* and for *A. parasiticus*, respectively Neither the temperature of the plasma jet or the treated hazelnuts was reported No quality evaluations conducted	[23]
Plasma jet	15 g peanuts	*A. flavus and A. niger*	Working gas: Argon Power: 200 W Rotary speed: 1000 rpm Flow rate: 30 L/min Treatment time: 3.5 min	Reduced to undetectable level from the initial inoculation level of 6.39 log CFU/g for *A. flavus* and 5.83 log CFU/g for *A. niger*, respectively Neither the temperature of the plasma jet or the treated peanuts was reported No effect on quality characteristics	[58]
Plasma jet	1 g peeled peanuts	*A. flavus*	Working gas: 9 L/min Argon mixed with 90 mL/min O_2_ Power: 20 W Treatment time: 10 min	100% microbial growth inhibition (initial inoculation: 50 µL of 5 × 10^9^ conidia/mL) Neither the temperature of the plasma jet or the treated peanuts was reported No quality evaluations conducted	[59]
DBD	4 g raw pistachio	*A. flavus*	Working gas: air Power: 130 W Voltage: 15 kV Treatment time: 3 min	4.0 log CFU/g reduction Pistachio kernels became darker Antioxidant activity of pistachio nuts slightly increased Neither the temperature of the plasma gas or the treated pistachio nuts was reported	[21]
DBD	Pistachio	*A. flavus*	Working gas: air Electric power consumption: 2.49 W/cm^3^ Voltage: 10.7 kV Treatment time: 18 min	*A. flavus* was completely removed after treatment Neither the temperature of the plasma gas or the treated pistachio nuts was reported No quality evaluations conducted	[22]
DBD	35 g hazelnuts	*A. flavus*	Working gas: Ambient air Synthetic air without water vapor Synthetic air with water vapor Flow rate: 3 L/min Power: 400 W Treatment time: 180 s	*A. flavus* reduction: Gas 1: 2.95 log CFU/g Gas 2: 2.56 log CFU/g Gas 3: below detection limit Neither the temperature of the plasma gas or the treated hazelnuts was reported No quality evaluations conducted	[24]
DBD	10 g raw groundnuts	*A. flavus* and *A. parasiticus*	Working gas: air Power: 60 W Voltage: 1950 V Treatment time: 24 min	99.9% and 99.5% inactivation of *A. paraticus* and *A. flavus*, respectively Complete disintegration of fungal spore membrane due to electroporation and etching caused by the reactive species of plasma Neither the temperature of the plasma gas or the treated groundnuts was reported No quality evaluations conducted	[60]
DBD	10 g raw peanut kernels	*A. flavus*	Working gas: 80% humidity air Voltage: 90 kV Treatment time: 10 min Post-treatment storage: 24 h	99.9% reduction (initial level of 3.50 log spore per sample) of *A. flavus* The temperature of the gas and treated peanut was lower than 42 °C No significant effect on quality	[57]

**Table 3 toxins-17-00129-t003:** Application of atmospheric cold plasma for aflatoxin degradation in nuts.

Plasma Source	Sample Characteristics	Treatment Conditions	Detection Technique	Significant Findings	References
Plasma jet	5 g hazelnut (3 µg/kg AFB_1_)	Working gas: air Flow rate: 3000 L/h of Air Power: 655 W Frequency: 25 kHz Treatment time: 1.7 min	HPLC	72% AFB1 reduction Maintained the sensory attributes of food evaluated by a sensory panel Neither the temperature of the plasma gas or the treated hazelnuts was reported	[26]
Plasma jet	10 pistachio kernels (20 µg/kg aflatoxins)	Voltage: 20 kV Working gas: 62%argon-38%air Flow rate: 3 L/min gas Treatment time: 15 min	Not avaliable	80% AFB1, 76% AFB2, 42% AFG1, and 51% AFG2 reduction No effect on quality characteristics of pistachio kernels After treatment the sample temperature was 27 °C	[20]
Atmospheric pressure plasma jet (APPJ)	15 g raw peanuts	Working gas: air Gas flow rate: 107 L/min Voltage: 4.4 kV Frequency: 70–90 kHz Treatment time: 2 min	ELISA	23% total aflatoxins reduction (from 62.3 to 48.2 ppb) Less bitterness than controls No changes in the quality of peanut oil	[62]
DBD	10 g groundnuts	Working gas: air Power: 60 W Treatment time: 12 min	HPLC	96.8% decrease in AFB1 production from *A. flavus* Upto 95% decrease in AFB1 from *A. paraticus* Neither the temperature of the plasma gas or the treated groundnuts was reported No quality assessment conducted	[23]
DBD	20 ng/g of AFB_1_ sprayed on 40 g raw hazelnuts	Working gas: N2 Flow rate:120 L/min Power: 1000 W Treatment time: 12 min	HPLC-MS/MS	Total aflatoxins and AFB1 reductions over 70% AFB1 and AFG1 more sensitive to plasma treatments compared to AFB2 and AFG2, respectively AFB1 more sensitive than AFG1 Neither the temperature of the plasma gas or the treated hazelnuts was reported No quality evaluations conducted	[27]
DBD	4 g of unskinned and milled pistachio sample was contaminated with AFB_1_ (383.19 µg/kg)	Working gas: air Power: 130 W Voltage: 15 kV Frequency: 20 kHz Treatment time: 60, 120, and 180 s	HPLC	AFB1 reduction: o 60 s: 32.31% o 120 s: 44.77% o 180 s: 52.42% Plasma treatment led to darker pistachio samples; slightly increased the antioxidant activity of pistachio samples	[21]
DBD	60 μL of 50 μg/mL AFB_1_ stock solution spot-inoculated on 15 g raw peanut kernels	Working gas: 5, 40, and 80% humidity air Voltage: 90 kV Treatment time: 2, 5, and 10 min Post-treatment storage: 0 h and 24 h	ELISA	71.3% AFB1 reduction achieved with a treatment of 2 min and 80% RH without post-treatment Reduction of AFB1 increased with increasing treatment time, air humidity and post-treatment storage The temperature of gas and treated peanuts was lower than 42 °C	[57]

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
