# Peer review of "A Review of Decontamination of Aspergillus spp. and Aflatoxin Control for Grains and Nuts with Atmospheric Cold Plasma"

_toxins, 2025, doi:10.3390/toxins17030129_

Round 1
Reviewer 1 Report
Comments and Suggestions for Authors
The review focuses on an interesting technology that has been tested for fungal and mycotoxin decontamination of foods. Nonetheless, the ideas are presented in a disorganized way, and a significant part of the manuscript is out of scope (in particular the first 11 pages).
Section 1- The introduction is a group of sentences that are not properly integrated. There is no flow in the subjects that are herein introduced.
Section 2 - The article lacks a proper frame to the core subject of decontamination. Instead of disserting on the characterization of Aspergillus and on aflatoxins (section 2), it would be more interesting to briefly discuss the methods and technologies that have been tested for decontaminations, focusing on their limitations, so as to justify the significance and interest of the atmospheric cold plasma technology.
Section 3, although with potential interest to the aims of the article, is also organized in a way that deviates from the article's scope.
Section 4 is not at all necessary, since it does not relate to the subject.
Only sections 5 and 6 are appropriate to the aims of the review. Even so, they also need improvement in terms of organization. Table 5 should be in section 5, not section 6. Section 6 subdivided into grains and nuts, but tables 3 and 4 are divided in Aspergillus and aflatoxins. It is confusing. There should also be more information on the limitations of the technology.
Sub-section 6.3 is not related with section 6, and should constitute an independent section (section 7).
Comments on the Quality of English LanguageMany improvements are required in the quality of english language. There are also many typos to be corrected.
Reviewer 2 Report
Comments and Suggestions for Authors
The presented manuscript gives an overview on Aspergillus spp.as well as an overview on detection and Aflatoxins Control for Grains and Nuts with Atmospheric Cold Plasma.
ACP only a small part of the whole review. Only two out of 6 deal with plasma. Consider changing the title to move the focus away from ACP.
There are flaws in the Introduction and some other parts of the manuscript. Please sharpen the text and concentrate the needed informatio, expand the text (see below for more suggestions).
I don´t understand the introduction. Terms and information are presented while I was wondering when Aspergillus or Aflatoxins are defined, which is not done before line 87.
Lines 66 to 73 make absolutely no sense. Starting with some picked out studies on Plasma decontamination without even introducing any relevant side- info is redundant. Rather give information on the traditional methods for inactivation and food processing and why the industry needs another method like ACP.
1. Introduction
The text does not introduce Aspergillus and Aflatoxins properly, e.g. line 33 starts with Aflatoxins without introduction and definition, Same for line 36 fungal diseases- Aspergillus has not been introduced as fungus ((class, family). Define both shortly.
Line 58: Factors for Aflatoxin production are presented but the main purpose for toxin production not definition of the toxins have been presented.
Line 63: how doe these aflatoxins variants differ- make square cross reverence to chapter 2
Line 81, “in “ is in capital letter in the middle of the sentence.
2. Characterization of Aspergillus spp. and Chemistry of Aflatoxins
Add biological reason for production of Aflatoxins .
Line 104: name all four classes.
Line 160 Consider to add sub-heading like Pathology including all infos on outbreaks and effect on human/animal health.
Line 167: Sentence is not finished, preposition missing.
Line 170 ff: This subchapter sounds like it has been written by someone else. Also, it disturbs the reading flow. Move to the start off the chapter. Or in front of line 160.
Table 1: Abreviations need to be deinfed (AFB, Afs)
I am missing a chapter after Chapter 4 on conventional methods to remove Aflatoxins and Aspergillus. Please add a short summary.
Line 244: what is an outstanding sensitivity? Define outstanding. I guess the presented methods vary in their sensitivity and detection limit.
Table 2: are there any infos on the respective detection limit and range of detection?
Line 392: Table 5 is mentioned before Table 3 and 4. Change order.
Line 394: Please add reference for “30.5 s were classified as short half-life species”
Sub-chapter 6.3- To my mind this should be an own chapter including also which of the presented configurations are the most promising for upscaling and industry application and what are challenges- e.g. legal permissions for ACP in food and feed?
Also discuss the presented result in Chapter 5/6 with respect to working gas and standardization/comparability/ how much sample can be treated in what time.
Lines 660 ff: Add discussion of user safety (Compliance with the regulations for air, water), energy, cost effectiveness compared to standard methods.
Line 643/44: Add reference.
Line 682, Capitalize first letter in Aspergillus
Comments on the Quality of English LanguageEnglish Language is fine for most of the parts. Some minor revisions should be made- see detailed comments.
Author Response
Please the attachment

Reviewer 3 Report
Comments and Suggestions for Authors
General
Much of the review is spent on extraneous material. Page 4-11 does not contain the word ‘plasma’ or the abbreviation ‘ACP’. The text runs 25 pages in length: The introduction is 3 pages and not very concise. The next 8 pages is only tangentially relevant to the subject of the review. The matter of ACP-based decontamination is really only addressed in section 6 of this review.
Rewrite with much closer focus on the stated purpose / objective. Reconsider use of the word “decontaminate” and carefully describe how you are using that word in this context. Describe HOW ACP is interacting with the fungus and with the mycotoxin.
Minor / specific points:
Inconsistent use of “aflatoxin” and “aflatoxins”. Line 9 has “aflatoxins contamination” and line 33 has “aflatoxin infestation”.
Line 43: Do you mean 38% of the global losses is in sub-Saharan Africa or 38% of the global loss is in developing countries?
Line 44: What do you mean “mainly”? ONLY fungi can produce mycotoxins. That’s why they are called MYCOtoxins.
Line 45: Wordy. instead of “…any stage of the crop production chain…” just say “…any stage of crop production…”
Line 49: I don’t care what the original publication said, it is laughable to say “36.96%.” Four significant digits suggests a level of precision that is not possible. It would probably be more realistic to report that the observed incidence ranged from below 40% to over 90%.
Line 54-57: This is a relevant citation, but seems really out of place in this paragraph.
Line 58: What kind of ‘injury’ are you talking about here?
Line 60-61: Beginning the sentence with “Thus” suggests some sort of connection to the previous statement, but I don’t see any connection. The previous statement is about host plant stress increasing the susceptibility to A. flavus infection. Then you say ‘thus, reduction of the mold is important’. The two statements have nothing to do with each other. I don’t even know what ‘reduction of the mold’ really means in this context.
Line 58-65: This paragraph has 5 sentences that have no connection to each other. Sentence 1 is useful, but probably needs slightly reworded. Sentence 2 makes no sense. Sentence 3 is fine, but just seemed stuffed in here without any context. Sentence 4-5 are connected to each other, but not with the rest of the paragraph.
Line 65: I don’t think the M aflatoxins are made by any fungi. They are metabolites when the other aflatoxins are processed by mammals.
Line 66: This wording is unclear. When you use ‘decontaminate’ in this context I don’t know if it means ‘kill the fungus’ or it means ‘remove the fungus’. The processes you describe here are designed to kill the fungus or render it biologically inactive. If you mean ‘kill’ why not just say ‘kill’? If ‘decontaminate’ means something more, explain what it does mean.
Line 69: What is “W” here?
Line 70-71: You say “another study” but you don’t give the citation.
Line 66-85: This paragraph makes no sense – the first half is about ‘decontamination’ and the second half goes back to describing the aflatoxin problem in the developing world.
Line 90: I have never seen the word “saprobic”. I’m guessing you mean “saprophytic”.
Line 91: why even say “…color, shape, and size of conidial head are important indicators…” if you aren’t going to say what the color, shape, and size are.
Line 97: The fungus is “…light to deep yellow…” or the spores are “…light to deep yellow…”
Line 105: A series of problems with the term “nontoxic”. First, the fungus isn’t really toxic. It PRODUCES a toxin, so the term you probably mean in “nontoxigenic”, i.e., not toxin-producing. But even that term is problematic because some of these strains that do not produce aflatoxin produce other mycotoxins, so they are not truly “nontoxigenic”, they are, more accurately, “nonaflatoxigenic”.
Line 105-106: Who says there are more spores in the air than in soil? How would you even compare air and soil concentrations? It’s not unusual to see greater than 1000 CFU in a cubic centimeter of soil and I can’t believe you would find greater than 1000 CFU in a cubic centimeter of air.
Section 2.2 This information has been reported and reviewed MANY times. In the context of this review, it is unacceptable that there is no discussion of how the aflatoxins react with / are deactivated by ACP.
Section 3: Why talk about the regulatory environment for aflatoxin in an ACP review unless you’re going to talk about how ACP can or cannot lead to regulatory compliance?
Section 4: I don’t know why this section is in this review. What is the connection between detection methods and ACP? This section is also quite superficial and possibly misleading. For example, what does “toxin manipulation” mean (Table 2, ELISA). To say that TLC requires “extensive sample treatments” is silly – for aflatoxin you can just spot crude extracts on a TLC plate. What do you mean FT-NIR “requires more time for calibration.” I’ve seen HPLC methods that have 2 hours worth of calibration.
Line 373: Was there a time when ‘consumers’ (what some would call ‘people’) didn’t like food to taste fresh?
Line 414: What does it mean that there was “complete removal” of A. flavus? You mean, “no detectable living A. flavus”?
Line 421: How meaningful is it to remove aflatoxin from the surface of the kernels? How much aflatoxin is on the surface compared to the interior of the kernel or to the bulk grain mass?
Table 3 is odd. The table seems quite comprehensive and gives MANY details, but not a single example of a quantitative change in aflatoxin concentration. Inactivation of A. flavus can be useful, but if there is not measurable change in in aflatoxin then it doesn’t seem very worthwhile.
Line 603: This sentence seems out of place. It is like something that should have been on the first page, not the 21st page.
Round 2
Reviewer 1 Report
Comments and Suggestions for Authors
The work has greatly improved and is now with quality to be published.
Please consider some minor revisions in the attached file.

There are some minor issues in need of revision in terms of the English quality.
Author Response
Manuscript has revised according to the minor comments.
Reviewer 3 Report
Comments and Suggestions for Authors
The authors satisfactorily addressed the issues of the previous draft.
Author Response
Manuscripts have revised according to the minor comments.